# On the Equivalence between Online and Private Learnability beyond Binary Classification

**Young Hun Jung**∗
Department of Statistics
University of Michigan
Ann Arbor, MI 48109
yhjung@umich.edu

**Baekjin Kim**∗
Department of Statistics
University of Michigan
Ann Arbor, MI 48109
baekjin@umich.edu

**Ambuj Tewari**
Department of Statistics
University of Michigan
Ann Arbor, MI 48109
tewaria@umich.edu

## Abstract

Alon et al. [4] and Bun et al. [10] recently showed that online learnability and private PAC learnability are equivalent in binary classification. We investigate whether this equivalence extends to multi-class classification and regression. First, we show that private learnability implies online learnability in both settings. Our extension involves studying a novel variant of the Littlestone dimension that depends on a tolerance parameter and on an appropriate generalization of the concept of threshold functions beyond binary classification. Second, we show that while online learnability continues to imply private learnability in multi-class classification, current proof techniques encounter significant hurdles in the regression setting. While the equivalence for regression remains open, we provide non-trivial sufficient conditions for an online learnable class to also be privately learnable.

## 1 Introduction

*Online learning* and *differentially-private (DP) learning* have been well-studied in the machine learning literature. While these two subjects are seemingly unrelated, recent papers have revealed a strong connection between online and private learnability via the notion of *stability* [2, 3, 17]. The notion of differential privacy is, at its core, less about privacy and more about algorithmic stability since the output distribution of a DP algorithm should be robust to small changes in the input. Stability also plays a key role in developing online learning algorithms such as follow-the-perturbed-leader (FTPL) and follow-the-regularized-leader (FTRL) [1].

Recently Alon et al. [4] and Bun et al. [10] showed that online learnability and private PAC learnability are equivalent in binary classification. Alon et al. [4] showed that private PAC learnability implies finite Littlestone dimension (Ldim) in two steps; (i) every approximately DP learner for a class with Ldim $d$ requires $\Omega(\log^* d)$ thresholds (see Section 2.4 for the definition of $\log^*$), and (ii) the class of thresholds over $\mathbb{N}$ cannot be learned in a private manner. Bun et al. [10] proved the converse statement via a notion of algorithmic stability, called *global stability*. They showed (i) every class with finite Ldim can be learned by a globally-stable learning algorithm and (ii) they use global stability to derive a DP algorithm. In this work, we investigate whether this equivalence extends to multi-class classification (MC) and regression, which is one of open questions raised by Bun et al. [10].

In general, online learning and private learning for MC and regression have been less studied. In binary classification without considering privacy, the Vapnik-Chervonenkis dimension (VCdim) of hypothesis classes yields tight sample complexity bounds in the batch learning setting, and Littlestone [20] defined Ldim as a combinatorial parameter that was later shown to fully characterize hypothesis classes that are learnable in the online setting [8]. Until recently, however, it was unknown what

---

∗Equal Contribution

complexity measures for MC or regression classes characterize online or private learnability. Daniely et al. [11] extended the Ldim to the MC setting, and Rakhlin et al. [22] proposed the sequential fat-shattering dimension, an online counterpart of the fat-shattering dimension in the batch setting [6].

## 1.1 Related works

DP has been extensively studied in the machine learning literature [12, 14, 23]. Private PAC and agnostic learning were formally studied in the seminal work of Kasiviswanathan et al. [18], and the sample complexities of private learners were characterized in the later work of Beimel et al. [7].

Dwork et al. [14] identified stability as a common factor of learning and differential privacy. Abernethy et al. [2] proposed a DP-inspired stability-based methodology to design online learning algorithms with excellent theoretical guarantees, and Agarwal and Singh [3] showed that stabilization techniques such as regularization or perturbation in online learning preserve DP. Feldman and Xiao [16] relied on communication complexity to show that every purely DP learnable class has a finite Ldim. Purely DP learnability is a stronger condition than online learnability, which means that there exist online learnable classes that are not purely DP learnable. More recently, Alon et al. [4] and Bun et al. [10] established the equivalence between online and private learnability in a non-constructive manner. Gonen et al. [17] derived an efficient black-box reduction from purely DP learning to online learning. In the paper we will focus on approximate DP instead of pure DP (see Definition 2).

## 1.2 Main results and techniques

Our main technical contributions are as follows.

- In Section 3, we develop a novel variant of the Littlestone dimension that depends on a tolerance parameter $\tau$, denoted by $\mathrm{Ldim}_\tau$. While online learnable regression problems do not naturally reduce to learnable MC problems by discretization, this relaxed complexity measure bridges online MC learnability and regression learnability in that it allows us to consider a regression problem as a relatively simpler MC problem (see Proposition 5).

- In Section 4, we show that private PAC learnability implies online learnability in both MC and regression settings. We appropriately generalize the concept of threshold functions beyond the binary classification setting and lower bound the number of these functions using the complexity measures (see Theorem 8). Then the argument of Alon et al. [4] that an infinite class of thresholds cannot be privately learned can be extended to both settings of interest.

- In Section 5, we show that while online learnability continues to imply private learnability in MC (see Theorem 11), current proof techniques based on *global stability* and *stable histogram* encounter significant obstacles in the regression problem. While this direction for regression setting still remains open, we provide non-trivial sufficient conditions for an online learnable class to also be privately learnable (see Theorem 15).

## 2 Preliminaries

We study multi-class classification and regression problems in this paper. In multi-class classification problems with $K \geq 2$ classes, we let $\mathcal{X}$ be the input space and $\mathcal{Y} = [K] \triangleq \{1, 2, \cdots, K\}$ be the output space, and the *standard zero-one loss* $\ell^{0-1}(\hat{y}; y) = \mathbb{I}(\hat{y} \neq y)$ is considered.

The regression problem is similar to the classification problem, except that the label becomes continuous, $\mathcal{Y} = [-1, 1]$, and the goal is to learn a real-valued function $f : \mathcal{X} \to \mathcal{Y}$ that approximates well labels of future instances. We consider the *absolute loss* $\ell^{abs}(\hat{y}; y) = |\hat{y} - y|$ in this setting. Results under the absolute loss can be generalized to any other Lipschitz losses with modified rates.

## 2.1 PAC learning

Let $\mathcal{X}$ be an input space, $\mathcal{Y}$ be an output space, and $\mathcal{D}$ be an unknown distribution over $\mathcal{X} \times \mathcal{Y}$. A *hypothesis* is a function mapping from $\mathcal{X}$ to $\mathcal{Y}$. The *population loss* of a hypothesis $h : \mathcal{X} \to \mathcal{Y}$ with respect to a loss function $\ell$ is defined by $\mathrm{loss}_{\mathcal{D}}(h) = \mathbb{E}_{(x,y) \sim \mathcal{D}}\big[\ell\big(h(x); y\big)\big]$. We also define the *empirical loss* of a hypothesis $h$ with respect to a loss function $\ell$ and a sample $S = \big((x_i, y_i)\big)_{1:n}$

as $\text{loss}_S(h) = \frac{1}{n}\sum_{i=1}^{n}\ell\big(h(x_i); y_i\big)$. The distribution $\mathcal{D}$ is said to be *realizable* with respect to $\mathcal{H}$ if there exists $h^\star \in \mathcal{H}$ such that $\text{loss}_{\mathcal{D}}(h^\star) = 0$.

**Definition 1** (PAC learning). *A hypothesis class $\mathcal{H}$ is PAC learnable with sample complexity $m(\alpha, \beta)$ if there exists an algorithm $\mathcal{A}$ such that for any $\mathcal{H}$-realizable distribution $\mathcal{D}$ over $\mathcal{X} \times \mathcal{Y}$, an accuracy and confidence parameters $\alpha, \beta \in (0,1)$, if $\mathcal{A}$ is given input samples $S = \big((x_i, y_i)\big)_{1:m} \sim \mathcal{D}^m$ such that $m \geq m(\alpha, \beta)$, then it outputs a hypothesis $h : \mathcal{X} \to \mathcal{Y}$ satisfying $\text{loss}_{\mathcal{D}}(h) \leq \alpha$ with probability at least $1 - \beta$. A learner which always returns hypotheses inside the class $\mathcal{H}$ is called a proper learner, otherwise is called an improper learner.*

## 2.2 Differential privacy

*Differential privacy* (DP) [14], a standard notion of statistical data privacy, was introduced to study data analysis mechanism that do not reveal too much information on any single sample in a dataset.

**Definition 2** (Differential privacy [14]). *Data samples $S, S' \in (\mathcal{X} \times \mathcal{Y})^n$ are called neighboring if they differ by exactly one example. A randomized algorithm $\mathcal{A} : (\mathcal{X} \times \mathcal{Y})^n \to \mathcal{Y}^{\mathcal{X}}$ is $(\epsilon, \delta)$-differentially private if for all neighboring data samples $S, S' \in (\mathcal{X} \times \mathcal{Y})^n$, and for all measurable sets $T$ of outputs,*

$$\mathbb{P}\big(\mathcal{A}(S) \in T\big) \leq e^\epsilon \cdot \mathbb{P}\big(\mathcal{A}(S') \in T\big) + \delta.$$

*The probability is taken over the randomness of $\mathcal{A}$. When $\delta = 0$ we say that $\mathcal{A}$ preserves pure differential privacy, otherwise (when $\delta > 0$) we say that $\mathcal{A}$ preserves approximate differential privacy.*

Combining the requirements of PAC and DP learnability yields the definition of private PAC learner.

**Definition 3** (Private PAC learning [18]). *A hypothesis class $\mathcal{H}$ is $(\epsilon, \delta)$-differentially private PAC learnable with sample complexity $m(\alpha, \beta)$ if it is PAC learnable with sample complexity $m(\alpha, \beta)$ by an algorithm $\mathcal{A}$ which is $(\epsilon, \delta)$-differentially private.*

## 2.3 Online learning

The online learning problem can be viewed as a repeated game between a learner and an adversary. Let $T$ be a time horizon and $\mathcal{H} \subset \mathcal{Y}^{\mathcal{X}}$ be a class of predictors over a domain $\mathcal{X}$. At time $t$, the adversary chooses a pair $(x_t, y_t) \in \mathcal{X} \times \mathcal{Y}$, and the learner observes the instance $x_t$, predicts a label $\hat{y}_t \in \mathcal{Y}$, and finally observes the loss $\ell\big(\hat{y}_t; y_t\big)$. This work considers the *full-information setting* where the learner receives the true label information $y_t$. The goal is to minimize the *regret*, namely the cumulative loss that the learner actually observed compared to the best prediction in hindsight:

$$\sum_{t=1}^{T} \ell\big(\hat{y}_t; y_t\big) - \min_{h^\star \in \mathcal{H}} \sum_{t=1}^{T} \ell\big(h^\star(x_t); y_t\big).$$

A class $\mathcal{H}$ is *online learnable* if for every $T$, there is an algorithm that achieves sub-linear regret $o(T)$ against any sequence of $T$ instances.

The *Littlestone dimension* is a combinatorial parameter that exactly characterizes online learnability for binary hypothesis classes [8, 20]. Daniely et al. [11] further extended this to the multi-class setting. We need the notion of mistake trees to define this complexity measure. A *mistake tree* is a binary tree whose internal nodes are labeled by elements of $\mathcal{X}$. Given a node $x$, its descending edges are labeled by distinct $k, k' \in \mathcal{Y}$. Then any root-to-leaf path can be expressed as a sequence of instances $\big((x_i, y_i)\big)_{1:d}$, where $x_i$ represents the $i$-th internal node in the path, and $y_i$ is the label of its descending edge in the path. We say that a tree $T$ is *shattered* by $\mathcal{H}$ if for any root-to-leaf path $\big((x_i, y_i)\big)_{1:d}$ of $T$, there is $h \in \mathcal{H}$ such that $h(x_i) = y_i$ for all $i \leq d$. The Littlestone dimension of multi-class hypothesis class $\mathcal{H}$, $\text{Ldim}(\mathcal{H})$, is the maximal depth of any $\mathcal{H}$-shattered mistake tree. Just like binary classification, a set of MC hypotheses $\mathcal{H}$ is online learnable if and only if $\text{Ldim}(\mathcal{H})$ is finite.

The (sequential) *fat-shattering dimension* is the scale-sensitive complexity measure for real-valued function classes [22]. A mistake tree for real-valued function class $\mathcal{F}$ is a binary tree whose internal nodes are labeled by $(x, s) \in \mathcal{X} \times \mathcal{Y}$, where $s$ is called a *witness to shattering*. Any root-to-leaf path in a mistake tree can be expressed as a sequence of tuples $\big((x_i, \epsilon_i)\big)_{1:d}$, where $x_i$ is the label of the

$i$-th internal node in the path, and $\epsilon_i = +1$ if the $(i+1)$-th node is the right child of the $i$-th node, and otherwise $\epsilon_i = -1$ (for the leaf node, $\epsilon_d$ can take either value). A tree $T$ is $\gamma$-shattered by $\mathcal{F}$ if for any root-to-leaf path $\big((x_i, \epsilon_i)\big)_{1:d}$ of $T$, there exists $f \in \mathcal{F}$ such that $\epsilon_i\,(f(x_i) - s_i) \geq \gamma/2$ for all $i \leq d$. The fat-shattering dimension at scale $\gamma$, denoted by $\mathrm{fat}_\gamma(\mathcal{F})$, is the largest $d$ such that $\mathcal{F}$ $\gamma$-shatters a mistake tree of depth $d$. For any function class $\mathcal{F} \subset [-1,1]^{\mathcal{X}}$, $\mathcal{F}$ is online learnable in the supervised setting under the absolute loss if and only if $\mathrm{fat}_\gamma(\mathcal{F})$ is finite for any $\gamma > 0$ [22].

The (sequential) *Pollard pseudo-dimension* is a scale-free fat-shattering dimension for real-valued function classes. For every $f \in \mathcal{F}$, we define a binary function $B_f : \mathcal{X} \times \mathcal{Y} \to \{-1, +1\}$ by $B_f(x, s) = \mathrm{sign}\,(f(x) - s)$ and let $\mathcal{F}^+ = \{B_f \mid f \in \mathcal{F}\}$. Then we define the Pollard pseudo-dimension by $\mathrm{Pdim}(\mathcal{F}) = \mathrm{Ldim}(\mathcal{F}^+)$. It is easy to check that $\mathrm{fat}_\gamma(\mathcal{F}) \leq \mathrm{Pdim}(\mathcal{F})$ for all $\gamma$. That being said, finite Pollard pseudo-dimension is a sufficient condition for online learnability but not a necessary condition (e.g., bounded Lipschitz functions on [0,1] separate the two notions).

### 2.4 Additional notation

We define a few functions in a recursive manner. The *tower function* $\mathrm{twr}_t$ and the *iterated logarithm* $\log^{(m)}$ are defined respectively as

$$\mathrm{twr}_t(x) = \begin{cases} x & \text{if } t = 0, \\ 2^{\mathrm{twr}_{t-1}(x)} & \text{if } t > 0, \end{cases} \quad \log^{(m)} x = \begin{cases} \log x & \text{if } m = 1, \\ \log^{(m-1)} \log x & \text{if } m > 1. \end{cases}$$

Lastly, we use $\log^* x$ to denote the minimal number of recursions for the iterated logarithm to return the value less than or equal to one:

$$\log^* x = \begin{cases} 0 & \text{if } x \leq 1, \\ 1 + \log^* \log x & \text{if } x > 1. \end{cases}$$

## 3 A link between multi-class and regression problems

As a tool to analyze regression problems, we discretize the continuous space $\mathcal{Y}$ into intervals and consider the problem as a multi-class problem. Specifically, given a function $f \in [-1,1]^{\mathcal{X}}$ and a scalar $\gamma$, we split the interval $[-1,1]$ into $\lceil \frac{2}{\gamma} \rceil$ intervals of length $\gamma$ and define $[f]_\gamma(x)$ to be the index of interval that $f(x)$ belongs to. We can also define $[\mathcal{F}]_\gamma = \{[f]_\gamma \mid f \in \mathcal{F}\}$. In this way, if the multi-class problem associated with $[\mathcal{F}]_\gamma$ is learnable, we can infer that the original regression problem is learnable up to accuracy $O(\gamma)$. Quite interestingly, however, the fact that $\mathcal{F}$ is (regression) learnable does not imply that $[\mathcal{F}]_\gamma$ is (multi-class) learnable. For example, it is well known that a class $\mathcal{F}$ of bounded Lipschitz functions on [0,1] is learnable, but $[\mathcal{F}]_1$ includes all binary functions on $[0,1]$, which is not online learnable.

In order to tackle this issue, we propose a generalized zero-one loss in multi-class problems. In particular, we define a *zero-one loss with tolerance $\tau$*,

$$\ell_\tau^{0-1}(\hat{y}; y) = \mathbb{I}(|y - \hat{y}| > \tau).$$

Note that the classical zero-one loss is simply $\ell_0^{0-1}$. This generalized loss allows the learner to predict labels that are not equal to the true label but close to it. This property is well-suited in our setting since as far as $|y - \hat{y}|$ is small, the absolute loss in the regression problem remains small.

We also extend the Littlestone dimension with tolerance $\tau$. Fix a tolerance level $\tau$. When we construct a mistake tree $T$, we add another constraint that each node's descending edges are labeled by two labels $k, k' \in [K]$ such that $\ell_\tau^{0-1}(k; k') = 1$. Let $\mathrm{Ldim}_\tau(\mathcal{H})$ be the maximal height of such binary shattered trees. (Again, $\mathrm{Ldim}_0(\mathcal{H})$ becomes the standard $\mathrm{Ldim}(\mathcal{H})$.)

We record several useful observations. The proofs can be found in Appendix A.

**Lemma 4.** *Let $\mathcal{H} \subset [K]^{\mathcal{X}}$ be a class of multi-class hypotheses.*

1. $\mathrm{Ldim}_\tau(\mathcal{H})$ *is decreasing in $\tau$.*

2. $\mathrm{SOA}_\tau$ *(Algorithm 1) makes at most $\mathrm{Ldim}_\tau(\mathcal{H})$ mistakes with respect to $\ell_\tau^{0-1}$.*

---
**Algorithm 1** Standard optimal algorithm with tolerance $\tau$ (SOA$_\tau$)
---
1: **Initialize:** $V_0 = \mathcal{H}$
2: **for** $t = 1, \cdots, T$ **do**
3:     Receive $x_t$
4:     For $k \in [K]$, let $V_t^{(k)} = \{h \in V_{t-1} \mid h(x_t) = k\}$
5:     Predict $\hat{y}_t = \arg\max_k \mathrm{Ldim}_\tau(V_t^{(k)})$
6:     Receive true label $y_t$ and update $V_t = V_t^{(y_t)}$
7: **end for**
---

*3. For any deterministic learning algorithm, an adversary can force* $\mathrm{Ldim}_{2\tau}(\mathcal{H})$ *mistakes with respect to* $\ell_\tau^{0-1}$.

Equipped with the relaxed loss, the following proposition connects regression learnability to multi-class learnability with discretization. We emphasize that even though the regression learnability does not imply multi-class learnability with the standard zero-one loss, learnability under $\ell_\tau^{0-1}$ can be derived. In addition to that, it can be shown that finite $\mathrm{Ldim}_\tau([\mathcal{F}]_\gamma)$ implies finite $\mathrm{fat}_\gamma(\mathcal{F})$.

**Proposition 5.** *Let $\mathcal{F} \subset [-1, 1]^{\mathcal{X}}$ be a regression hypothesis class and suppose $\mathrm{fat}_\gamma(\mathcal{F}) = d$. Then we have for any positive integer $n$,*

$$\mathrm{Ldim}_n([\mathcal{F}]_{\gamma/2(n+1)}) \geq d \geq \mathrm{Ldim}_n([\mathcal{F}]_{\gamma/n}).$$

*Proof.* Since $\mathrm{fat}_\gamma(\mathcal{F}) = d$, in the online learning setting an adversary can force any deterministic learner to suffer at least $\gamma/2$ absolute loss for $d$ rounds. If we think of this problem as a multi-class classification problem using the hypothesis class $[\mathcal{F}]_{\gamma/2(n+1)}$, using the same strategy, the adversary can force any deterministic learner to make mistakes with respect to $\ell_n^{0-1}$ for $d$ rounds. Note that the adversary reveals less information to the learner in the discretized multi-class problem. Then Lemma 4 implies $\mathrm{Ldim}_n([\mathcal{F}]_{\gamma/2(n+1)}) \geq d$.

On the other hand, suppose $\mathrm{Ldim}_n([\mathcal{F}]_{\gamma/n}) > d$ and let $T$ be the binary shattered tree with tolerance $n$. For each node, we can set the witness point to be the middle point between the two labels of descending edges, and the resulting tree is $\gamma$-shattered by $\mathcal{F}$. This contradicts the fact that $\mathrm{fat}_\gamma(\mathcal{F}) = d$, and hence we obtain $d \geq \mathrm{Ldim}_n([\mathcal{F}]_{\gamma/n})$. $\qquad\square$

There exist a few works that used regression models in multi-class classification [21, 24]. To the best of our knowledge, however, our work is the first one that studies regression learnability by transforming the problem into a discretized classification problem along with a novel bridge, *Littlestone dimension with tolerance*.

## 4   Private learnability implies online learnability

In this section, we show that if a class of functions is privately learnable, then it is online learnable. To do so, we prove a lower bound of the sample complexity of privately learning algorithms using either $\mathrm{Ldim}(\mathcal{H})$ for the multi-class hypotheses or $\mathrm{fat}_\gamma(\mathcal{F})$ for the regression hypotheses. Alon et al. [4] proved this in the binary classification setting first by showing that any large Ldim class contains sufficiently many threshold functions and then providing a lower bound of the sample complexity to privately learn threshold functions. We adopt their arguments, but one of the first non-trivial tasks is to define analogues of threshold functions in multi-class or regression problems. Note that, a priori, it is not clear what the right analogy is. Let us first introduce threshold functions in the binary case. We say a binary hypothesis class $\mathcal{H}$ has $n$ thresholds if there exist $\{x_i\}_{1:n} \subset \mathcal{X}$ and $\{h_i\}_{1:n} \subset \mathcal{H}$ such that $h_i(x_j) = 1$ if $i \leq j$ and $h_i(x_j) = 0$ if $i > j$. We extend this as below.

**Definition 6** (Threshold functions in multi-class problems). *Let $\mathcal{H} \subset [K]^{\mathcal{X}}$ be a hypothesis class. We say $\mathcal{H}$ contains $n$ thresholds with a gap $\tau$ if there exist $k, k' \in [K]$, $\{x_i\}_{1:n} \subset \mathcal{X}$, and $\{h_i\}_{1:n} \subset \mathcal{H}$ such that $|k - k'| > \tau$ and $h_i(x_j) = k$ if $i \leq j$ and $h_i(x_j) = k'$ if $i > j$.*

**Definition 7** (Threshold functions in regression problems). *Let $\mathcal{F} \subset [-1, 1]^{\mathcal{X}}$ be a hypothesis class. We say $\mathcal{F}$ contains $n$ thresholds with a margin $\gamma$ if there exist $\{x_i\}_{1:n} \subset \mathcal{X}$, $\{f_i\}_{1:n} \subset \mathcal{F}$, and $u, u' \in [-1, 1]$ such that $|u - u'| \geq \gamma$ and $|f_i(x_j) - u| \leq \frac{\gamma}{20}$ if $i \leq j$ and $|f_i(x_j) - u'| \leq \frac{\gamma}{20}$ if $i > j$.*

---

**Algorithm 2** COLORANDCHOOSE

---

1: **Input:** multi-class hypothesis class $\mathcal{H} \subset [K]^{\mathcal{X}}$, shattered binary tree $T$, tolerance $\tau$
2: Choose an arbitrary hypothesis $h_0 \in \mathcal{H}$
3: Color each vertex $x$ of $T$ by $h_0(x) \in [K]$
4: Find a color $k$ such that the sub-tree $T' \subset T$ of color $k$ has the largest height
5: Let $x_0$ be the root node of $T'$
6: Let $x_1$ be a child of $x_0$ such that the edge $(x_0, x_1)$ is labeled as $k'$ with $|k - k'| > \frac{\tau}{2}$
7: Let $T''$ be a sub-tree of $T'$ rooted at $x_1$
8: Let $\mathcal{H}' = \{h \in \mathcal{H} \mid h(x_0) = k'\}$
9: **Output:** $k, k', h_0, x_0, \mathcal{H}', T''$

---

In Definition 7, we allow the functions to oscillate with a margin $\frac{\gamma}{20}$ which is arbitrary. Any small margin compared to $|u - u'|$ would work, but this number is chosen to facilitate later arguments.

Next we show that complex hypothesis classes contain a sufficiently large set of threshold functions. The following theorem extends the results by Alon et al. [4, Theorem 3]. A complete proof can be found in Appendix B.

**Theorem 8** (Existence of a large set of thresholds). *Let $\mathcal{H} \subset [K]^{\mathcal{X}}$ and $\mathcal{F} \subset [-1, 1]^{\mathcal{X}}$ be multi-class and regression hypothesis classes, respectively.*

1. *If $\mathrm{Ldim}_{2\tau}(\mathcal{H}) \geq d$, then $\mathcal{H}$ contains $\lfloor \frac{\log_K d}{K^2} \rfloor$ thresholds with a gap $\tau$.*

2. *If $\mathrm{fat}_\gamma(\mathcal{F}) \geq d$, then $\mathcal{F}$ contains $\lfloor \frac{\gamma^2}{10^4} \log_{100/\gamma} d \rfloor$ thresholds with a margin $\frac{\gamma}{5}$.*

*Proof sketch.* We begin with the multi-class setting. Suppose $d = K^{K^2 t}$. It suffices to show $\mathcal{H}$ contains $t$ thresholds. Let $T$ be a shattered binary tree of height $d$ and tolerance $2\tau$. Letting $\mathcal{H}_0 = \mathcal{H}$ and $T_0 = T$, we iteratively apply COLORANDCHOOSE (Algorithm 2). Namely, we write

$$k_n, k'_n, h_n, x_n, \mathcal{H}_n, T_n = \text{COLORANDCHOOSE}(\mathcal{H}_{n-1}, T_{n-1}, 2\tau). \tag{1}$$

Observe that for all $n$, we can infer $h_n(x_n) = h_n(x) = k_n$ for all internal vertices $x$ of $T_n$ ($\because$ line 4 of Algorithm 2) and $h(x_n) = k'_n$ for all $h \in \mathcal{H}_n$ ($\because$ line 8 of Algorithm 2).

Additionally, it can be shown that the height of $T_n$ is no less than $\frac{1}{K}$ times the height of $T_{n-1}$ (see Lemma 16 in Appendix B). This means that the iterative step (1) can be repeated $K^2 t$ times since $d = K^{K^2 t}$. Then there exist $k, k'$ and indices $\{n_i\}_{i=1}^t$ such that $k_{n_i} = k$ and $k'_{n_i} = k'$ for all $i$.

It is not hard to check that the functions $\{h_{n_i}\}_{1:t}$ and the arguments $\{x_{n_i}\}_{1:t}$ form thresholds with labels $k, k'$. Since $|k - k'| > \tau$ ($\because$ line 6 of Algorithm 2), this completes the proof.

The result in the regression setting can also be shown in a similar manner using Proposition 5.  □

Alon et al. [4, Theorem 1] proved a lower bound of the sample complexity in order to privately learn threshold functions. Then the multi-class result (with $\tau = 0$) of Theorem 8 immediately implies that if $\mathcal{H}$ is privately learnable, then it is online learnable. For the regression case, we need to slightly modify the argument to deal with the margin condition in Definition 7. The next theorem summarizes the result, and the proof appears in Appendix B.

**Theorem 9** (Lower bound of the sample complexity to privately learn thresholds). *Let $\mathcal{F} = \{f_i\}_{1:n} \subset [-1, 1]^{\mathcal{X}}$ be a set of threshold functions with a margin $\gamma$ on a domain $\{x_i\}_{1:n} \subset \mathcal{X}$ along with bounds $u, u' \in [-1, 1]$. Suppose $\mathcal{A}$ is a $(\frac{\gamma}{200}, \frac{\gamma}{200})$-accurate learning algorithm for $\mathcal{F}$ with sample complexity $m$. If $\mathcal{A}$ is $(\epsilon, \delta)$-DP with $\epsilon = 0.1$ and $\delta = O(\frac{1}{m^2 \log m})$, then it can be shown that $m \geq \Omega(\log^* n)$.*

Combining Theorem 8 and 9, we present our main result.

**Corollary 10** (Private learnability implies online learnability). *Let $\mathcal{H} \subset [K]^{\mathcal{X}}$ and $\mathcal{F} \subset [-1, 1]^{\mathcal{X}}$ be multi-class and regression hypothesis classes, respectively. Let $\mathrm{Ldim}(\mathcal{H}) = \mathrm{fat}_\gamma(\mathcal{F}) = d$. Suppose there is a learning algorithm $\mathcal{A}$ that is $(\frac{1}{16}, \frac{1}{16})$-accurate for $\mathcal{H}$ ($(\frac{\gamma}{200}, \frac{\gamma}{200})$-accurate for $\mathcal{F}$) with sample complexity $m$. If $\mathcal{A}$ is $(\epsilon, \delta)$-DP with $\epsilon = 0.1$ and $\delta = O(\frac{1}{m^2 \log m})$, then $m \geq \Omega(\log^* d)$.*

# 5 Online learnability implies private learnability

In this section, we show that online-learnable multi-class hypothesis classes can be learned in a DP manner. For regression hypothesis classes, we provide sufficient conditions for private learnability.

## 5.1 Multi-class classification

Bun et al. [10] proved that every binary hypothesis class with a finite Ldim is privately learnable by introducing a new notion of algorithmic stability called *global stability* as an intermediate property between online learnability and differentially-private learnability. Their arguments can be naturally extended to MC hypothesis classes, which is summarized in the next theorem.

**Theorem 11** (Online MC learning implies private MC learning). *Let $\mathcal{H} \subset [K]^{\mathcal{X}}$ be a MC hypothesis class with $\mathrm{Ldim}(\mathcal{H}) = d$. Let $\epsilon, \delta \in (0, 1)$ be privacy parameters and let $\alpha, \beta \in (0, 1/2)$ be accuracy parameters. For $n = O_d\left(\frac{\log(1/\beta\delta)}{\alpha\epsilon}\right)$, there exists an $(\epsilon, \delta)$-DP learning algorithm such that for every realizable distribution $\mathcal{D}$, given an input sample $S \sim \mathcal{D}^n$, the output hypothesis $f = \mathcal{A}(S)$ satisfies $\mathrm{loss}_{\mathcal{D}}(f) \leq \alpha$ with probability at least $1 - \beta$.*

While we consider the realizable setting in Theorem 11, a similar result also holds in the agnostic setting. The extension to the agnostic setting is discussed in Appendix C.3 due to limited space.

As a key to the proof of Theorem 11, we introduce global stability (GS) as follows.

**Definition 12** (Global stability [10]). *Let $n \in \mathbb{N}$ be a sample size and $\eta > 0$ be a global stability parameter. An algorithm $\mathcal{A}$ is $(n, \eta)$-GS with respect to $\mathcal{D}$ if there exists a hypothesis $h$ such that $\mathbb{P}_{S \sim \mathcal{D}^n}\left(\mathcal{A}(S) = h\right) \geq \eta$.*

Theorem 11 can be proved in two steps. We first show that every MC hypothesis class with a finite Ldim is learnable by a GS algorithm $\mathcal{A}$ (Theorem 13). Then we prove that any GS algorithm can be extended to a DP learning algorithm with a finite sample complexity.

**Theorem 13** (Online MC learning implies GS learning). *Let $\mathcal{H} \subset [K]^{\mathcal{X}}$ be a MC hypothesis class with $\mathrm{Ldim}(\mathcal{H}) = d$. Let $\alpha > 0$, and $m = \left((4K)^{d+1} + 1\right) \times \lceil\frac{d\log K}{\alpha}\rceil$. Then there exists a randomized algorithm $G : (\mathcal{X} \times [K])^m \to [K]^{\mathcal{X}}$ such that for a realizable distribution $\mathcal{D}$ and an input sample $S \sim \mathcal{D}^m$, there exists a $h$ such that*

$$\mathbb{P}\left(G(S) = h\right) \geq \frac{K-1}{(d+1)K^{d+1}} \quad and \quad \mathrm{loss}_{\mathcal{D}}(h) \leq \alpha.$$

Next, we give a brief overview on how to construct a GS learner $G$ and a DP learner $M$ in order to prove Theorem 11. The complete proofs are deferred to Appendix C.

### 5.1.1 Online multi-class learning implies globally-stable learning

Let $\mathcal{H}$ be a MC hypothesis class with $\mathrm{Ldim}(\mathcal{H}) = d$ and $\mathcal{D}$ be a realizable distribution over examples $\left(x, c(x)\right)$ where $c \in \mathcal{H}$ is an unknown target hypothesis. Recall that $\mathcal{H}$ is learnable by $\mathrm{SOA}_0$ (Algorithm 1) with at most $d$ mistakes on any realizable sequence. Prior to building a GS learner $G$, we construct a distribution $\mathcal{D}_k$ by appending $k$ *tournament examples* between random samples from $\mathcal{D}$, which force $\mathrm{SOA}_0$ to make at least $k$ mistakes when run on $S$ drawn from $\mathcal{D}_k$. Using the fact that $\mathrm{SOA}_0$ identifies the true labeling function after making $d$ mistakes, we can show that there exists $k \leq d$ and a hypothesis $f : \mathcal{X} \to [K]$ such that

$$\mathbb{P}_{S \sim \mathcal{D}_k, T \sim \mathcal{D}^n}\left(\mathrm{SOA}_0(S \circ T) = f\right) \geq K^{-d}.$$

A GS learner $G$ is built by firstly drawing $k \in \{0, 1, \cdots, d\}$ uniformly at random and then running the $\mathrm{SOA}_0$ on $S \circ T$ where $S \sim \mathcal{D}_k, T \sim \mathcal{D}^n$. The learner $G$ outputs a good hypothesis that enjoys small population loss with probability at least $\frac{K^{-d}}{d+1}$. We defer the detailed construction of $\mathcal{D}_k$ and proofs to Appendix C.

### 5.1.2 Globally-stable learning implies private multi-class learning

Let $G$ be a $(\eta, m)$-GS algorithm with respect to a target distribution $\mathcal{D}$. We run $G$ on $k$ independent samples of size $m$ to non-privately produce a long list $H := (h_i)_{1:k}$. The *Stable Histogram* algorithm

is a primary tool that allows us to publish a short list of frequent hypotheses in a DP manner. The fact that $G$ is GS ensures that some good hypotheses appear frequently in $H$. Then Lemma 14 implies that these good hypotheses remain in the short list with high probability. Once we obtain a short list, a generic DP learning algorithm [18] is applied to privately select an accurate hypothesis.

**Lemma 14** (Stable Histogram [13, 19]). *Let $X$ be any data domain. For $n \geq O(\frac{\log(1/\eta\beta\delta)}{\eta\epsilon})$, there exists an $(\epsilon, \delta)$-DP algorithm* HIST *which with probability at least $1-\beta$, on input $S = (x_i)_{1:n}$ outputs a list $L \subset X$ and a sequence of estimates $a \in [0,1]^{|L|}$ such that (i) every $x$ with $\mathrm{Freq}_S(x) \geq \eta$ appears in $L$, and (ii) for every $x \in L$, the estimate $a_x$ satisfies $|a_x - \mathrm{Freq}_S(x)| \leq \eta$ where $\mathrm{Freq}_S(x) := \big|\{i \in [n] \mid x_i = x\}\big|/n$.*

### 5.2 Regression

In classification, *Global Stability* was an essential intermediate property between online and private learnability. A natural approach to obtaining a DP algorithm from an online-learnable real-valued function class $\mathcal{F}$ is to transform the problem into a multi-class problem with $[\mathcal{F}]_\gamma$ for some $\gamma$ and then construct a GS learner using the previous techniques. If $[\mathcal{F}]_\gamma$ is privately-learnable, then we can infer that the original regression problem is also private-learnable up to an accuracy $O(\gamma)$.

Unfortunately, however, finite $\mathrm{fat}_\gamma(\mathcal{F})$ only implies finite $\mathrm{Ldim}_1([\mathcal{F}]_\gamma)$, and $\mathrm{Ldim}([\mathcal{F}]_\gamma)$ can still be infinite (see Proposition 5). This forces us to run $\mathrm{SOA}_1$ instead of $\mathrm{SOA}_0$, and as a consequence, after making $\mathrm{Ldim}_1([\mathcal{F}]_\gamma)$ mistakes, the algorithm can identify the true function up to some tolerance. Therefore we only get the relaxed version of GS property as follows; there exist $k \leq d$ and a hypothesis $f : \mathcal{X} \to [K]$ such that

$$\mathbb{P}_{S \sim \mathcal{D}_k, T \sim \mathcal{D}^n}\big(\mathrm{SOA}_1(S \circ T) \approx_1 f\big) \geq (\gamma/2)^d$$

where $f \approx_1 g$ means $\sup_{x \in \mathcal{X}} \big|f(x) - g(x)\big| \leq 1$. If we proceed with this relaxed condition, it is no longer guaranteed the long list $H$ contains a good hypothesis with sufficiently high frequency. This hinders us from using Lemma 14, and a private learner cannot be produced in this manner. The limitation of proving the equivalence in regression stems from existing proof techniques. With another method, it is still possible to show that online-learnable real-valued function classes can be learned by a DP algorithm. Instead, we provide sufficient conditions for private learnability in regression problems.

**Theorem 15** (Sufficient conditions for private regression learnability). *Let $\mathcal{F} \subset \mathcal{Y}^\mathcal{X}$ be a real-valued function class such that $\mathrm{fat}_\gamma(\mathcal{F}) < \infty$ for every $\gamma > 0$. If one of the following conditions holds, then $\mathcal{F}$ is privately learnable.*

1. *Either $\mathcal{F}$ or $\mathcal{X}$ is finite.*

2. *The range of $\mathcal{F}$ over $\mathcal{X}$ is finite (i.e., $\big|\{f(x) \mid f \in \mathcal{F}, x \in \mathcal{X}\}\big| < \infty$).*

3. *$\mathcal{F}$ has a finite cover with respect to the sup-norm at every scale.*

4. *$\mathcal{F}$ has a finite sequential Pollard Pseudo-dimension.*

We present the proof of Condition 4, and proofs of other conditions are deferred to Appendix C.4.

*Proof of Condition 4.* Assume for contradiction that there exists $\gamma$ such that $\mathrm{Ldim}([\mathcal{F}]_\gamma) = \infty$. Then we can obtain a shattered tree $T$ of an arbitrary depth. Choose an arbitrary node $x$. Note that its descending edges are labeled by $k, k' \in [[2/\gamma]]$. We can always find a witness to shattering $s$ between the intervals corresponding to $k$ and $k'$. With these witness values, the tree $T$ must be zero-shattered by $\mathcal{F}$. Since the depth of $T$ can be arbitrarily large, this contradicts to $\mathrm{Pdim}(\mathcal{F})$ being finite. From this, we can claim that $\mathrm{Ldim}([\mathcal{F}]_\gamma) \leq \mathrm{Pdim}(\mathcal{F})$ for any $\gamma$. Then using the ideas in Section 5.1, we can conclude that $[\mathcal{F}]_\gamma$ is private-learnable for any $\gamma$. Therefore the original class $\mathcal{F}$ is also private-learnable. $\square$

We emphasize that Conditions 3 and 4 do not imply each other. For example, a class of point functions $\mathcal{F}^{\mathrm{point}} := \{\mathbb{I}(\cdot = x) \mid x \in \mathcal{X}\}$ does not have a finite sup-norm cover because any two distinct functions have the sup-norm difference one, but $\mathrm{Pdim}(\mathcal{F}^{\mathrm{point}}) = 1$. A class $\mathcal{F}^{\mathrm{Lip}}$ of bounded Lipschitz functions on $[0, 1]$ has an infinite sequential Pollard pseudo-dimension, but $\mathcal{F}^{\mathrm{Lip}}$ has a finite cover with respect to the sup-norm due to compactness of $[0, 1]$ along with the Lipschitz property.

# 6 Discussion

We have pushed the study of the equivalence between online and private learnability beyond binary classification. We proved that private learnability implies online learnability in the MC and regression settings. We also showed the converse in the MC setting and provided sufficient conditions for an online learnable class to also be privately learnable in regression problems.

We conclude with a few suggestions for future work. First, we need to understand whether online learnability implies private learnability in the regression setting. Second, like [10], we create an improper DP learner for an online learnable class. It would be interesting to see if we can construct *proper* DP learners. Third, Gonen et al. [17] provide an efficient black-box reduction from *pure* DP learning to online learning. It is natural to explore whether such efficient reductions are possible for *approximate* DP algorithms for MC and regression problems. Finally, there are huge gaps between the lower and upper bounds for sample complexities in both classification and regression settings. It would be desirable to show tighter bounds and reduce these gaps.

## Broader Impact

As this paper is purely theoretical, discussing broader impact is not applicable.

## Acknowledgments and Disclosure of Funding

We acknowledge the support of NSF via grants CAREER IIS-1452099 and IIS-2007055.

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
