[Supplementary Material]

# A   Section 3 details

We prove Lemma 4.

**Lemma 4** (restated). *Let $\mathcal{H} \subset [K]^{\mathcal{X}}$ be a class of multi-class hypotheses.*

1. $\mathrm{Ldim}_\tau(\mathcal{H})$ *is decreasing in* $\tau$.

2. $\mathrm{SOA}_\tau$ *(Algorithm 1) makes at most* $\mathrm{Ldim}_\tau(\mathcal{H})$ *mistakes with respect to* $\ell_\tau^{0-1}$.

3. *For any deterministic learning algorithm, an adversary can force* $\mathrm{Ldim}_{2\tau}(\mathcal{H})$ *mistakes with respect to* $\ell_\tau^{0-1}$.

*Proof.* Part 1 follows by observing that if $T$ is a binary shattered tree with tolerance $\tau$, then so is it with tolerance $\tau' < \tau$.

For part 2, assume $\mathrm{SOA}_\tau$ makes a mistake at round $t$. We claim that $\mathrm{Ldim}_\tau(V_{t+1}) < \mathrm{Ldim}_\tau(V_t)$. If $\mathrm{Ldim}_\tau$ does not decrease, we can infer that

$$\mathrm{Ldim}_\tau(V_t^{(\hat{y}_t)}) = \mathrm{Ldim}_\tau(V_t^{(y_t)}) = \mathrm{Ldim}_\tau(V_t) =: d.$$

Then we can find binary trees $T_1$ and $T_2$ of height $d$ that are shattered by $V_t^{(\hat{y}_t)}$ and $V_t^{(y_t)}$, respectively. By concatenating $T_1$ and $T_2$ with a root node $x_t$ and its edges labeled by $\hat{y}_t$ and $y_t$, we can obtain a binary tree $T$ of height $d+1$ that is shattered by $V_t$. This contradicts to $\mathrm{Ldim}_\tau(V_t) = d$ and proves our assertion.

To prove part 3, let $T$ be a binary shattered tree of height $\mathrm{Ldim}_{2\tau}(\mathcal{H})$. For a given node $x$, suppose the adversary shows $x$ to the learner. Since the descending edges have labels apart from each other by more than $2\tau$, the adversary can choose a label that incurs a mistake with respect to $\ell_\tau^{0-1}$. Thus by following down the tree $T$ from the root node, the adversary can force $\mathrm{Ldim}_{2\tau}(\mathcal{H})$ mistakes.   □

# B   Section 4 details

In this section, the proofs omitted in Section 4 are presented.

## B.1   Proof of Theorem 8

We first define *sub-trees*. Let $T$ be a binary tree. Any node of $T$ becomes its sub-tree of height 1. For $h > 1$, choose a node $x$ and let $T_1$ and $T_2$ be the trees that are rooted at its two children. A sub-tree of height $h$ is obtained by aggregating a sub-tree of height $h-1$ of $T_1$ and a sub-tree of height $h-1$ of $T_2$ at the root node $x$. Note that if the original tree $T$ is shattered by some hypothesis class, then so is any sub-tree of it.

Next we prove a helper lemma.

**Lemma 16.** *Suppose there are $n$ colors $C = \{c_i\}_{1:n}$ and $n$ positive integers $\{d_i\}_{1:n}$. Let $T$ be a binary tree of height $-(n-1) + \sum_{i=1}^n d_i$ whose vertices are colored by $C$. Then there exists a color $c_i$ such that $T$ has a sub-tree of height $d_i$ in which all internal vertices are colored by $c_i$.*

*Proof.* We will prove by induction on $\sum_{i=1}^n d_i$. If $d_i = 1$ for all $i$, then the height of $T$ becomes 1, and the statement holds trivially. Now suppose the lemma holds for any $d_i$'s whose summation is less than $N$ and let $T$ have the height $N - n + 1$. Without loss of generality, we may assume that the root node $x_0$ is colored by $c_1$. We consider two sub-trees $T_1, T_2$ of height $N - n$ whose root nodes are children of $x_0$. Let $e_1 = d_1 - 1$ and $e_i = d_i$ for $i > 1$. Since $\sum_{i=1}^n e_i = N - 1$, by the inductive assumption each $T_j$ has a sub-tree of height $e_{i_j}$ in which all internal vertices are colored by $c_{i_j}$. If $i_j \neq 1$ for some $j$, then we are done because $e_{i_j} = d_{i_j}$. If $i_j = 1$ for all $j = 1, 2$, then merging these two trees with the node $x_0$ forms a sub-tree of height $e_1 + 1 = d_1$ of color $c_1$. This completes the inductive argument.   □

Now we are ready to prove Theorem 8.

**Theorem 8** (restated). *Let $\mathcal{H} \subset [K]^{\mathcal{X}}$ and $\mathcal{F} \subset [-1,1]^{\mathcal{X}}$ be multi-class and regression hypothesis classes, respectively.*

1. If $\mathrm{Ldim}_{2\tau}(\mathcal{H}) \geq d$, then $\mathcal{H}$ contains $\lfloor \frac{\log_K d}{K^2} \rfloor$ thresholds with a gap $\tau$.

2. If $\mathrm{fat}_\gamma(\mathcal{F}) \geq d$, then $\mathcal{F}$ contains $\lfloor \frac{\gamma^2}{10^4} \log_{100/\gamma} d \rfloor$ thresholds with a margin $\frac{\gamma}{5}$.

*Proof.* We begin with the multi-class setting. Suppose $d = K^{K^2 t}$. It suffices to show $\mathcal{H}$ contains $t$ thresholds. Let $T$ be a shattered binary tree of height $d$ and tolerance $2\tau$. Letting $\mathcal{H}_0 = \mathcal{H}$ and $T_0 = T$, we iteratively apply COLORANDCHOOSE (Algorithm 2). Namely, we write

$$k_n, k'_n, h_n, x_n, \mathcal{H}_n, T_n = \text{COLORANDCHOOSE}(\mathcal{H}_{n-1}, T_{n-1}, 2\tau). \tag{2}$$

Observe that for all $n$, we can infer $h_n(x_n) = h_n(x) = k_n$ for all internal vertices $x$ of $T_n$ ($\because$ line 4 of Algorithm 2) and $h(x_n) = k'_n$ for all $h \in \mathcal{H}_n$ ($\because$ line 8 of Algorithm 2).

Additionally, Lemma 16 ensures that the height of $T_n$ is no less than $\frac{1}{K}$ times the height of $T_{n-1}$. This means that the iterative step (2) can be repeated $K^2 t$ times since $d = K^{K^2 t}$. Then there exist $k, k'$ and indices $\{n_i\}_{i=1}^t$ such that $k_{n_i} = k$ and $k'_{n_i} = k'$ for all $i$.

It is not hard to check that the functions $\{h_{n_i}\}_{1:t}$ and the arguments $\{x_{n_i}\}_{1:t}$ form thresholds with labels $k, k'$. Since $|k - k'| > \tau$ ($\because$ line 6 of Algorithm 2), this completes the proof.

Now we move on to the regression setting. Proposition 5 implies that $\mathrm{Ldim}_{20}([\mathcal{F}]_{\gamma/50}) \geq \mathrm{Ldim}_{24}([\mathcal{F}]_{\gamma/50}) \geq d$. Then using the previous result in the multi-class setting, we can deduce that $[\mathcal{F}]_{\gamma/50}$ contains $n := \lfloor \frac{\gamma^2}{10^4} \log_{100/\gamma} d \rfloor$ thresholds with a gap 10. This means that there exist $k, k' \in [\frac{100}{\gamma}]$, $\{x_i\}_{1:n} \subset \mathcal{X}$, and $\{[f_i]_{\gamma/50}\}_{1:n} \subset \mathcal{H}$ such that $|k - k'| \geq 10$ and

$$[f_i]_{\gamma/50}(x_j) = \begin{cases} k & \text{if } i \leq j \\ k' & \text{if } i > j \end{cases}.$$

Let $u, u'$ be the middles points of the intervals that correspond to the labels $k, k'$. Then it is easy to check that $|u - u'| \geq \gamma/5$ and

$$f_i(x_j) \in \begin{cases} [u - \frac{\gamma}{100}, u + \frac{\gamma}{100}) & \text{if } i \leq j \\ [u' - \frac{\gamma}{100}, u' + \frac{\gamma}{100}) & \text{if } i > j \end{cases}.$$

This proves the theorem. $\qquad\square$

### B.2 Proof of Theorem 9

**Theorem 9** (restated). *Let $\mathcal{F} = \{f_i\}_{1:n} \subset [-1, 1]^\mathcal{X}$ be a set of threshold functions with a margin $\gamma$ on a domain $\{x_i\}_{1:n} \subset \mathcal{X}$ along with bounds $u, u' \in [-1, 1]$. Suppose $\mathcal{A}$ is a $(\frac{\gamma}{200}, \frac{\gamma}{200})$-accurate learning algorithm for $\mathcal{F}$ with sample complexity $m$. If $\mathcal{A}$ is $(\epsilon, \delta)$-DP with $\epsilon = 0.1$ and $\delta = O(\frac{1}{m^2 \log m})$, then it can be shown that $m \geq \Omega(\log^* n)$.*

*Proof.* The proof consists of two main lemmas. Lemma 19 proves that there is a large homogeneous set (see Definition 17). Then Lemma 21 yields the lower bound of the sample complexity when there exists a large homogeneous set. In particular, from these two lemmas, we can deduce that

$$\frac{\log^{(m)} n}{2^{O(m \log m)}} \leq 2^{O(m^2 \log^{(2)} m)}.$$

This means that there exists a constant $c$ such that

$$\log^{(m)} n \leq e^{cm^2 \log m}.$$

Observing that $\log^* \left( \log^{(m)} n \right) \geq \left( \log^* n \right) - m$ and $\log^* \left( 2^{O(m^2 \log^{(2)} m)} \right) = O(\log^* m)$, we can check the desired inequality $m \geq \Omega(\log^* n)$. $\qquad\square$

### B.2.1 Existence of a large homogenous set

Suppose $\mathcal{A}$ is a learning algorithm over a finite domain $D$. The hypothesis class consists of threshold functions over $D$ with bounds $u, u'$. According to Definition 7, $u$ and $u'$ can be in an arbitrary order as long as $|u - u'| > \gamma$. But for simpler presentation, without loss of generality, we will assume $u > u'$. Also, let $\bar{u} = \frac{u+u'}{2}$. We define the following quantity:

$$\mathcal{A}_S(x) = \mathbb{P}_{f \sim \mathcal{A}(S)}\big(f(x) \geq \bar{u}\big).$$

The definition of homogenous sets (Definition 17) and Lemma 19 are adopted from Alon et al. [4]. Assume that $\mathcal{X}$ is linearly ordered. Given a training set $S = \big((x_i, y_i)\big)_{1:m}$, we say $S$ is *increasing* if $x_1 \leq \cdots \leq x_m$. Additionally, we say $S$ is *balanced* if $y_i = u'$ for all $i \leq \frac{m}{2}$ and $y_i = u$ for all $i > \frac{m}{2}$. Given $x \in \mathcal{X}$, we define $\mathrm{ord}_S(x) = \big|\{i \mid x_i \leq x\}\big|$. Lastly, we use $S_{\mathcal{X}}$ to denote $(x_i)_{1:m}$.

**Definition 17** (*m-homogeneous set*)**.** *A set $D' \subset D$ is m-homogeneous with respect to a learning algorithm $\mathcal{A}$ if there are numbers $p_i \in [0,1]$ for $0 \leq i \leq m$ such that for every increasing balanced sample $S \in (D' \times \{u, u'\})^m$ and for every $x \in D' \setminus S_{\mathcal{X}}$*

$$|\mathcal{A}_S(x) - p_i| \leq \frac{1}{100m},$$

*where $i = \mathrm{ord}_S(x)$.*

The following theorem is a well-known result in Ramsey theory. It was originally introduced by Erdos and Rado [15] and rephrased by Alon et al. [4].

**Theorem 18** (Alon et al. [4, Theorem 11])**.** *Let $s > t \geq 2$ and $q$ be integers, and let $N \geq \mathrm{twr}_t(3sq \log q)$. Then for every coloring of the subsets of size $t$ of a universe of size $N$ using $q$ colors, there is a homogeneous subset [2] of size $s$.*

The next lemma states that we can find a large homogeneous set.

**Lemma 19** (Existence of a large homogeneous set)**.** *Let $\mathcal{A}$ be a learning algorithm over a domain $D$ with $|D| = n$. Then there exists a set $D' \subset D$ which is m-homogeneous with respect to $\mathcal{A}$ such that*

$$|D'| \geq \frac{\log^{(m)} n}{2^{O(m \log m)}}.$$

*Proof.* We first define a coloring on the $(m+1)$-subsets of $D$. Let $B = \{x_1 < x_2 < \cdots < x_{m+1}\}$ be an $(m+1)$-subset. For each $i \in [m+1]$, let $B^{(i)} = B \setminus \{x_i\}$. Then by labeling the first half of $B^{(i)}$ by $u'$ and the second half by $u$, we get a balanced increasing training set $S^{(i)}$. Then we compute $p_i$ that is of the form $\frac{t}{100m}$ and closest to $\mathcal{A}_{S^{(i)}}(x_i)$ (in case of ties, choose the smaller one). Then we color $B$ by the tuple $(p_i)_{1:m+1}$.

This scheme includes $(100m + 1)^{m+1}$ colors, and Theorem 18 provides that there exists a set $D'$ of size larger than

$$\frac{\log^{(m)} n}{3(100m+1)^{m+1}(m+1)\log(100m+1)} = \frac{\log^{(m)} n}{2^{O(m \log m)}}$$

such that all $(m+1)$-subsets of $D'$ have the same color. It is easy to verify that this set is indeed $m$-homogeneous with respect to $\mathcal{A}$ according to Definition 17. $\qquad\square$

### B.2.2 Large homogeneous set implies the lower bound

Recall that PAC learning is defined with respect to $\mathrm{loss}_{\mathcal{D}}$ (see Definition 1). When $\mathrm{loss}_{\mathcal{D}}$ is replaced by $\mathrm{loss}_S$, we say an algorithm $\mathcal{A}$ *empirically learns* a training set $S$. Bun et al. [9, Lemma 5.9] prove that if a hypothesis class is PAC learnable, then there exists an empirical learner as well.

**Lemma 20** (Empirical learner)**.** *Suppose $\mathcal{A}$ is an $(\epsilon, \delta)$-DP PAC learner for a hypothesis class $\mathcal{H}$ that is $(\alpha, \beta)$-accurate and has sample complexity $m$. Then there is an $(\epsilon, \delta)$-DP and $(\alpha, \beta)$-accurate empirical learner for $\mathcal{H}$ with sample complexity $9m$.*

The next is the main lemma.

**Lemma 21** (Large homogeneous sets imply lower bounds on sample complexity). *Suppose a learning algorithm $\mathcal{A}$ is $(\epsilon, \delta)$-DP with sample complexity $m$. Let $X = [N]$ be $m$-homogeneous with respect to $\mathcal{A}$. If $\epsilon = 0.1$, $\delta \le \frac{1}{1000m^2 \log m}$, and $\mathcal{A}$ empirically learns the threshold functions with a margin $\gamma$ over $X$ with $(\frac{\gamma}{200}, \frac{\gamma}{200})$-accuracy, then*

$$N \le 2^{O(m^2 \log^{(2)} m)}.$$

*Proof.* The proof is done by combining Lemma 22 and Lemma 23, which come below. $\square$

This is the first helper lemma to prove Lemma 21. It adopts Alon et al. [4, Lemma 12].

**Lemma 22.** *Let $\mathcal{A}, X, m, N$ as in Lemma 21 and assume $N > 2m$. Then there exists a family $\mathcal{P} = \{P_i\}_{1:N-m}$ of distributions over $\{-1, 1\}^{N-m}$ that satisfies the following two properties.*

1. *$P_i$ and $P_j$ are $(\epsilon, \delta)$-indistinguishable for all $i \ne j$.*

2. *There exists $r \in [0, 1]$ such that for all $i, j \in [N - m]$,*

$$\mathbb{P}_{v \sim P_i}(v_j = 1) \begin{cases} \le r - \frac{1}{10m} & \text{if } j < i \\ \ge r + \frac{1}{10m} & \text{if } j > i \end{cases}.$$

*Proof.* Let $(p_i)_{0:m}$ be the probability list associated with $m$-homogeneous set $X = [N]$. We first prove that there exists $i^*$ such that $p_{i^*} - p_{i^*-1} \ge \frac{1}{4m}$. Fix an increasing balanced training set $S := ((x_i, y_i))_{1:m} \in (X \times \{u, u'\})^m$ such that $x_i - x_{i-1} \ge 2$ for all $i$, which is possible by the assumption $N > 2m$. By the definition of threshold functions with a margin $\gamma$, we can infer

$$\min_f \text{loss}_S(f) \le \frac{\gamma}{20} = 0.05\gamma,$$

where the minimum is taken over the threshold functions with a margin $\gamma$.

Furthermore, since $\mathcal{A}$ is an $(\alpha = \frac{\gamma}{200}, \beta = \frac{\gamma}{200})$-accurate empirical learner, we can bound the expected loss of $\mathcal{A}(S)$ as

$$\mathbb{E}_{f \sim \mathcal{A}(S)} \text{loss}_S(f) \le \alpha + \beta + \min_f \text{loss}_S(f) \le 0.06\gamma. \tag{3}$$

Also, we can lower bound the expected empirical loss by using the quantity $\mathcal{A}_S(x_i)$ as follows (recall that we assumed $u > u'$)

$$\mathbb{E}_{f \sim \mathcal{A}(S)} \text{loss}_S(h) \ge \frac{1}{m} \cdot \frac{\gamma}{2} \left( \sum_{i=1}^{m/2} [\mathcal{A}_S(x_i)] + \sum_{i=m/2+1}^{m} [1 - \mathcal{A}_S(x_i)] \right). \tag{4}$$

Combining (3) and (4), we can show that there exists $j \le \frac{m}{2}$ such that $\mathcal{A}_S(x_j) \le \frac{1}{4}$. Let $S' = (S \setminus \{(x_j, y_j)\}) \cup \{(x_j + 1, y_j)\}$. Since $\mathcal{A}$ is $(\epsilon = 0.1, \delta \le \frac{1}{1000m^2 \log m})$-DP, we have

$$p_{j-1} - \frac{1}{100m} \le \mathcal{A}_{S'}(x_j) \le \frac{1}{4} e^\epsilon + \delta \le 0.3,$$

which implies that $p_{j-1} \le 0.3 + \frac{1}{100m} \le \frac{1}{3}$. Similarly, we can find $k > \frac{m}{2}$ such that $p_{k+1} \ge \frac{2}{3}$. Then we can find $i^* \in [j, k+1]$ such that $p_{i^*} - p_{i^*-1} \ge \frac{1}{4m}$, which proves our assertion.

Now we construct $\mathcal{P} = \{P_i\}_{1:N-m}$. Given $i$, let

$$B^{(i)} = \{1, \cdots, i^* - 1\} \cup \{i^* + i\} \cup \{i^* + N - m + 1, \cdots, N\} \subset X.$$

Observe that $B^{(i)}$ and $B^{(j)}$ only differ by one item at the position $i^*$. Then define $S^{(i)}$ to be the balanced increasing training set built upon $B^{(i)}$. Given a hypothesis $f$, we can compute a $N - m$ dimensional binary vector $v \in \{-1, 1\}^{N-m}$ such that

$$v_j = \mathbb{I}\left(f(i^* - 1 + j) \ge \bar{u}\right), \text{ where } \bar{u} = \frac{u + u'}{2}.$$

This mapping induces a distribution over $\{-1, 1\}^{N-m}$ from $\mathcal{A}(S^{(i)})$, which we define to be $P_i$.

Due to DP property of $\mathcal{A}$, $P_i$ and $P_j$ are $(\epsilon, \delta)$-indistinguishable. Furthermore, our construction of $i^*$ ensures the second property with $r = \frac{p_{i-1} + p_i}{2}$. This completes the proof. $\square$

The second helper lemma is shown by Alon et al. [4, Lemma 13].

**Lemma 23.** *Suppose the family $\mathcal{P}$ as in Lemma 22 exists. Then $N - m \leq 2^{1000m^2 \log^{(2)} m}$.*

## C  Section 5 details

We provide details omitted in Section 5.

### C.1  Proof of Theorem 13

Let $\mathcal{H}$ be a multi-class hypothesis class with $\mathrm{Ldim}(\mathcal{H}) = d$ and $\mathcal{D}$ be a realizable distribution over examples $(x, c(x))$ where $c \in \mathcal{H}$ is an unknown target hypothesis. The globally-stable (GS) leaner $G$ for $\mathcal{H}$ will make use of the Standard Optimal Algorithm (SOA$_0$, Algorithm 1).

SOA$_0$ can be simply extended to non-realizable sequences as follows.

**Definition 24** (Extending the SOA$_0$ to non-realizable sequences)**.** *Consider a run of* SOA$_0$ *on examples $\big((x_i, y_i)\big)_{1:m}$, and let $h_t$ denote the predictor used by the* SOA$_0$ *after observing the first $t$ examples. Then after observing $(x_{t+1}, y_{t+1})$, proceed as below.*

- *If $\big((x_i, y_i)\big)_{1:t+1}$ is realizable by some $h \in \mathcal{H}$, then apply the usual update rule of the* SOA$_0$ *to obtain $h_{t+1}$.*

- *Else, set $h_{t+1}$ as $h_{t+1}(x_{t+1}) = y_{t+1}$, and $h_{t+1}(x) = h_t(x)$ for every $x \neq x_{t+1}$. That is to say, $h_{t+1}$ no longer belongs to $\mathcal{H}$.*

This update rule keeps updating the predictor $h_t$ to agree with the last example while observing the sequences which are not necessarily realized by a hypothesis in $\mathcal{H}$. Due to this extension, our resulting algorithm possibly becomes improper.

The finite Littlestone class is online learnable by SOA$_0$ (Algorithm 1) with at most $d$ mistakes on any realizable sequence. Prior to building a GS learner $G$, we define a distribution $\mathcal{D}_k$ as in Algorithm 3.

---

**Algorithm 3** Distribution $\mathcal{D}_k$
___

1: $\mathcal{D}_0$ : output an empty set with probability 1
2: Let $k \geq 1$. If there exists an $f$ satisfying $\mathbb{P}_{S \sim \mathcal{D}_{k-1}, T \sim \mathcal{D}^n}\big(\mathrm{SOA}_0(S \circ T) = f\big) \geq K^{-d}$,
   or if $\mathcal{D}_{k-1}$ is undefined, then $\mathcal{D}_k$ is undefined
3: Else, $\mathcal{D}_k$ is defined recursively as follows
4:   (i) Randomly sample $S_0, S_1 \sim \mathcal{D}_{k-1}$ and $T_0, T_1 \sim \mathcal{D}^n$
5:   (ii) Let $f_0 = \mathrm{SOA}_0(S_0 \circ T_0)$ and $f_1 = \mathrm{SOA}_0(S_1 \circ T_1)$
6:   (iii) If $f_0 = f_1$, go back to step (i)
7:   (iv) Else, pick $x \in \{x \mid f_0(x) \neq f_1(x)\}$ and sample $y \sim [K]$ uniformly at random
8:   (v) If $f_0(x) \neq y$, output $S_0 \circ T_0 \circ (x, y)$ and $S_1 \circ T_1 \circ (x, y)$ otherwise

---

Let $k$ be such that $\mathcal{D}_k$ is well-defined and consider a sample $S$ drawn from $\mathcal{D}_k$. The size of $\mathcal{D}_k$ is $k \cdot (n + 1)$, and they consist of $k \cdot n$ instances randomly drawn from $\mathcal{D}$ and $k$ examples generated in Item 3(iv) of Algorithm 3. We call these $k$ examples *tournament examples*. Due to the construction of $\mathcal{D}_k$, SOA$_0$ always errs in tournament rounds, which means that SOA$_0$ makes at least $k$ mistakes when run on $S \circ T$ where $S \sim \mathcal{D}_k, T \sim \mathcal{D}^n$.

A natural way to obtain a GS learning algorithm $G$ is to run the SOA$_0$ on this carefully chosen sample $S \circ T$. In fact, the output enjoys both global stability in multi-class learning and good generalization as follows.

**Lemma 25** (Global Stability)**.** *There exist $k \leq d$ and a hypothesis $f : \mathcal{X} \to [K]$ such that*

$$\mathbb{P}_{S \sim \mathcal{D}_k, T \sim \mathcal{D}^n}\big(\mathrm{SOA}_0(S \circ T) = f\big) \geq K^{-d}.$$

*Proof.* Assume for contradiction that $\mathcal{D}_d$ is well-defined and for every $f$,

$$\mathbb{P}_{S \sim \mathcal{D}_k, T \sim \mathcal{D}^n}\big(\mathrm{SOA}_0(S \circ T) = f\big) < K^{-d}.$$

In each tournament example $(x_i, y_i)$, the label $y_i$ is drawn uniformly at random from $[K]$. Accordingly, with probability $K^{-d}$ over $S \sim \mathcal{D}_d$, all $d$ tournament examples are consistent with the true labeling function $c$ and thus $S \circ T$ becomes consistent with $c$. Since the number of total mistakes of $\text{SOA}_0$ should be no more than $d$, we can deduce that $\text{SOA}_0(S \circ T) = c$. This implies that

$$\mathbb{P}_{S \sim \mathcal{D}_k, T \sim \mathcal{D}^n}\big(\text{SOA}_0(S \circ T) = c\big) \geq K^{-d},$$

which is a contradiction, and hence completes the proof. $\qquad \square$

**Lemma 26** (Generalization). *Let $k$ be such that $\mathcal{D}_k$ is well-defined. Then for every $f$ such that*

$$\mathbb{P}_{S \sim \mathcal{D}_k, T \sim \mathcal{D}^n}\big(\text{SOA}_0(S \circ T) = f\big) \geq K^{-d}$$

*satisfies* $loss_{\mathcal{D}}(f) \leq \frac{d \log K}{n}$.

*Proof.* Let $f$ be such hypothesis and let $\alpha = loss_{\mathcal{D}}(f)$. We will argue that $K^{-d} \leq (1-\alpha)^n$. Then the following result is derived, $\alpha \leq \frac{d \log K}{n}$ using the fact that $(1-\alpha)^n \leq e^{-n\alpha}$.

By the property of $\text{SOA}_0$, $\text{SOA}_0(S \circ T)$ is consistent with $T$. Thus, if $\text{SOA}_0(S \circ T) = f$, then it must be the case that $f$ is consistent with $T$. By assumption, $\text{SOA}_0(S \circ T) = f$ holds with probability at least $K^{-d}$ and $f$ is consistent with $T$ with probability $(1-\alpha)^n$ where $n$ is the size of $T$. This gives the desired inequality. $\qquad \square$

One challenge associated with the distribution $\mathcal{D}_k$ is computational limitation. It may require an unbounded number of samples from the target distribution $\mathcal{D}$, since during generation of tournament examples the number of samples drawn from $\mathcal{D}$ depends on how many times Item 3(i)-(iii) will be repeated. To handle this practical issue, we suggest a Monte-Carlo Variant of $\mathcal{D}_k$, $\tilde{\mathcal{D}}_k$, by setting an upper bound $N$ of random samples drawn from $\mathcal{D}$ as an input parameter. Algorithm 4 summarizes how we construct the distribution $\tilde{\mathcal{D}}_k$.

---

**Algorithm 4** Distribution $\tilde{\mathcal{D}}_k$

1: Let $n$ be the auxiliary sample size and $N$ be an upper bound on the number of samples from $\mathcal{D}$
2: $\tilde{\mathcal{D}}_0$ : output an empty set with probability 1
3: Let $k \geq 1$. $\tilde{\mathcal{D}}_k$ is defined recursively by the following processes
4:    ($\star$) Throughout the process, if more than $N$ examples are drawn from $\mathcal{D}$, then output "Fail"
5:    (i) Randomly sample $S_0, S_1 \sim \tilde{\mathcal{D}}_{k-1}$ and $T_0, T_1 \sim \mathcal{D}^n$
6:    (ii) Let $f_0 = \text{SOA}_0(S_0 \circ T_0)$ and $f_1 = \text{SOA}_0(S_1 \circ T_1)$
7:    (iii) If $f_0 = f_1$, go back to step (i)
8:    (iv) Else, pick $x \in \{x \mid f_0(x) \neq f_1(x)\}$ and sample $y \sim [K]$ uniformly at random
9:    (v) If $f_0(x) \neq y$, output $S_0 \circ T_0 \circ (x, y)$ and $S_1 \circ T_1 \circ (x, y)$ otherwise

---

The next step is to specify the upper bound $N$. The following lemma characterizes the expected sample complexity of sampling from $\mathcal{D}_k$.

**Lemma 27** (Expected sample complexity of sampling from $\mathcal{D}_k$). *Let $k$ be such that $\mathcal{D}_k$ is well-defined and $M_k$ be the number of samples from $\mathcal{D}$ when generating $S \sim \mathcal{D}_k$. Then we have $\mathbb{E} M_k \leq 4^{k+1} \cdot n$.*

*Proof.* Initially, $\mathbb{E} M_0 = 0$ since $\mathcal{D}_0$ outputs an empty set with probability 1. It suffices to show that for all $0 < i < k$, $\mathbb{E} M_{i+1} \leq 4\mathbb{E} M_i + 4n$ to conclude the desired inequality by induction.

Let $R$ be the number of times Item 3(i) was executed during generation of $S \sim \mathcal{D}_{i+1}$, and $R$ is distributed geometrically with a success probability $\theta$, where

$$\theta = 1 - \mathbb{P}_{S_0, S_1, T_0, T_1}\big(\text{SOA}_0(S_0 \circ T_0) = \text{SOA}_0(S_1 \circ T_1)\big)$$
$$= 1 - \sum_f \big(\mathbb{P}_{S,T}\big(\text{SOA}_0(S \circ T) = f\big)\big)^2$$
$$\geq 1 - K^{-d}.$$

The last inequality holds because $i < k$ and hence $\mathcal{D}_i$ is well-defined, which implies that $\mathbb{P}_{S,T}\big(\text{SOA}_0(S \circ T) = f\big) \leq K^{-d}$ for all $f$.

Let $M_{i+1}$ be a random variable expressed as $M_{i+1} = \sum_{j=1}^{\infty} M_{i+1}^{(j)}$ where

$$M_{i+1}^{(j)} = \begin{cases} 0, & \text{if } R < j \\ \text{the number of examples from } \mathcal{D} \text{ in the } j\text{-th execution of Item 3(i),} & \text{if } R \geq j \end{cases}.$$

Thus, we have

$$\mathbb{E}M_{i+1} = \sum_{j=1}^{\infty} \mathbb{E}M_{i+1}^{(j)} = \sum_{j=1}^{\infty}(1-\theta)^{j-1} \cdot (2\mathbb{E}M_i + 2n)$$

$$= \frac{1}{\theta} \cdot (2\mathbb{E}M_i + 2n) \leq 4\mathbb{E}M_i + 4n,$$

where the last inequality holds since $\theta \geq 1 - K^{-d} \geq 1/2$ since $K \geq 2$ and $d \geq 1$. $\qquad\square$

Equipped with Lemma 25,26, and 27, we are ready to prove Theorem 13.

**Theorem 13** (restated). *Let $\mathcal{H} \subset [K]^{\mathcal{X}}$ be a MC hypothesis class with $\mathrm{Ldim}(\mathcal{H}) = d$. Let $\alpha > 0$, and $m = \left((4K)^{d+1} + 1\right) \times \lceil \frac{d \log K}{\alpha} \rceil$. Then there exists a randomized algorithm $G : (\mathcal{X} \times [K])^m \to [K]^{\mathcal{X}}$ such that for a realizable distribution $\mathcal{D}$ and an input sample $S \sim \mathcal{D}^m$, there exists a $h$ such that*

$$\mathbb{P}\big(G(S) = h\big) \geq \frac{K-1}{(d+1)K^{d+1}} \quad and \quad loss_{\mathcal{D}}(h) \leq \alpha.$$

*Proof.* The globally-stable algorithm $G$ is defined in Algorithm 5.

---

**Algorithm 5** Algorithm $G$

---

1: **Input :** target distribution $\tilde{\mathcal{D}}_k$, auxiliary sample size $n = \lceil \frac{d \log K}{\alpha} \rceil$, and the sample complexity upper bound $N = (4K)^{d+1} \cdot n$
2: Draw $k \in \{0, 1, \cdots, d\}$ uniformly at random
3: **Output :** $h = \mathrm{SOA}_0(S \circ T)$, where $T \sim \mathcal{D}^n, S \sim \tilde{\mathcal{D}}_k$

---

The sample complexity of $G$ is $|S| + |T| \leq N + n = \left((4K)^{d+1} + 1\right) \times \lceil \frac{d \log K}{n} \rceil$. By Lemma 25 and 26, there exists $k^\star \leq d$ and $f^\star$ such that

$$\mathbb{P}_{S \sim \mathcal{D}_{k^\star}, T \sim \mathcal{D}^n}\big(\mathrm{SOA}(S \circ T) = f^\star\big) \geq \frac{1}{K^d}, \quad loss_{\mathcal{D}}(f^\star) \leq \frac{d \log K}{n} \leq \alpha.$$

Let $M_{k^\star}$ denote the number of random examples from $\mathcal{D}$ during generation of $S \sim \mathcal{D}_{k^\star}$. We obtain the following inequality from Lemma 27 and Markov's inequality,

$$\mathbb{P}\big(M_{k^\star} > (4K)^{d+1} \cdot n\big) \leq \mathbb{P}\big(M_{k^\star} > K^{d+1} \cdot 4^{k^\star+1} \cdot n\big) \leq K^{-(d+1)}.$$

Accordingly,

$$\mathbb{P}_{S \sim \tilde{\mathcal{D}}_{k^\star}, T \sim \mathcal{D}^n}\big(\mathrm{SOA}_0(S \circ T) = f^\star\big)$$
$$\geq \mathbb{P}_{S \sim \mathcal{D}_{k^\star}, T \sim \mathcal{D}^n}\big(\mathrm{SOA}_0(S \circ T) = f^\star \text{ and } M_{k^\star} \leq (4K)^{d+1} \cdot n\big)$$
$$\geq \mathbb{P}_{S \sim \mathcal{D}_{k^\star}, T \sim \mathcal{D}^n}\big(\mathrm{SOA}_0(S \circ T) = f^\star\big) - \mathbb{P}\big(M_{k^\star} > (4K)^{d+1} \cdot n\big)$$
$$\geq K^{-d} - K^{-(d+1)} = (K-1)K^{-(d+1)}$$

Since $k = k^\star$ with probability $\frac{1}{d+1}$, $G$ outputs $f^\star$ with probability at least $\frac{K-1}{(d+1)K^{d+1}}$. $\qquad\square$

### C.2 Globally-stable learning implies private multi-class learning

In this section, we utilize the GS algorithm from the previous section to derive a DP learning algorithm with a finite sample complexity. Theorem 11 establishes that online multi-class learnability implies private multi-class learnability, which can be proved by combining Theorem 13 and Theorem 28.

**Theorem 28** (Globally-stable learning implies private multi-class learning). *Let $\mathcal{H} \subset [K]^{\mathcal{X}}$ be a multi-class hypothesis class. Let $G : (\mathcal{X} \times [K])^m \to [K]^{\mathcal{X}}$ be a randomized algorithm such that for a realizable distribution $\mathcal{D}$ and $S \sim \mathcal{D}^m$, there exists a hypothesis $h$ such that $\mathbb{P}(G(S) = h) \geq \eta$ and $\mathrm{loss}_{\mathcal{D}}(h) \leq \alpha/2$. Then for some $n = O(\frac{m \log(1/\eta\beta\delta)}{\eta\epsilon} + \frac{\log(1/\eta\beta)}{\alpha\epsilon})$, there exists an $(\epsilon, \delta)$-DP algorithm $M$ which for $n$ i.i.d. samples from $\mathcal{D}$, outputs a hypothesis $\hat{h}$ such that $\mathrm{loss}_{\mathcal{D}}(\hat{h}) \leq \alpha$ with probability at least $1 - \beta$.*

To construct a private learner $M$, we first introduce standard tools in the DP community such as *Stable Histogram* and *Generic Private Learner*.

**Lemma 14** (Stable Histogram, restated). *Let $X$ be any data domain. For $n \geq O(\frac{\log(1/\eta\beta\delta)}{\eta\epsilon})$, there exists an $(\epsilon, \delta)$-DP algorithm $\mathrm{HIST}$ which with probability at least $1 - \beta$, on input $S = (x_1, \cdots, x_n)$ outputs a list $L \in X$ and a sequence of estimates $a \in [0,1]^{|L|}$ such that*

1. *Every $x$ with $\mathrm{Freq}_S(x) \geq \eta$ appears in $L$, and*

2. *For every $x \in L$, the estimate $a_x$ satisfies $|a_x - \mathrm{Freq}_S(x)| \leq \eta$,*

*where $\mathrm{Freq}_S(x) = |\{i \in [n] \mid x_i = x\}|/n$.*

**Lemma 29** (Generic Private Learner, [10]). *Let $\mathcal{H} \subset [K]^{\mathcal{X}}$ be a collection of multi-class hypotheses. For $n = O(\frac{\log|\mathcal{H}| + \log(1/\beta)}{\alpha\epsilon})$, there exists an $(\epsilon, 0)$-DP algorithm $\mathrm{GENERICLEARNER} : (\mathcal{X} \times [K])^n \to \mathcal{H}$ satisfying the following; let $\mathcal{D}$ be a distribution over $\mathcal{X} \times [K]$ such that there exists an $h^{\star} \in \mathcal{H}$ with $\mathrm{loss}_{\mathcal{D}}(h^{\star}) \leq \alpha$. Then on input $S \sim \mathcal{D}^n$, $\mathrm{GENERICLEARNER}$ outputs, with probability at least $1 - \beta$, a hypothesis $\hat{h} \in \mathcal{H}$ such that $\mathrm{loss}_S(\hat{h}) \leq 2\alpha$.*

Now we are ready to prove Theorem 28.

*Proof of Theorem 28.* The learning algorithm $M$ is built on top of the Stable Historgram and the Generic Private Learner as described in Algorithm 6. According to Lemma 14 and 29, we choose parameters

$$k = O\big(\frac{\log(1/\eta\beta\delta)}{\eta\epsilon}\big), \quad n' = O\big(\frac{\log(1/\eta\beta)}{\alpha\epsilon}\big).$$

---

**Algorithm 6** Differentially-Private Learner $M$

---

1: Let $S_1, \cdots, S_k$ each consist of i.i.d. samples of size $m$ from $\mathcal{D}$. Run $G$ on each batch of samples producing $h_1 = G(S_1), \cdots, h_k = G(S_k)$
2: Run the Stable Histogram algorithm $\mathrm{HIST}$ on input $H = (h_1, \cdots, h_k)$ using privacy $(\epsilon/2, \delta)$ and accuracy $(\eta/8, \beta/3)$, publishing a list $L$ of frequent hypotheses
3: Let $S'$ consist of $n'$ i.i.d. samples from $\mathcal{D}$. Run $\mathrm{GENERICLEARNER}(S')$ using $L$ with privacy $\epsilon/2$ and accuracy $(\alpha/2, \beta/3)$ to output a hypothesis $\hat{h}$

---

We show that the algorithm $M$ is $(\epsilon, \delta)$-DP. During the executions of $G(S_1), \cdots G(S_k)$, a change to one entry in a certain $S_i$ changes at most one outcome $h_i \in H$. Thus, differential privacy for this step is observed by taking expectations over the coin tosses of all the executions of $G$. Then the differential privacy for overall algorithm holds by simple composition of differentially-private $\mathrm{HIST}$ and $\mathrm{GENERICLEARNER}$.

Next, we prove that the algorithm $M$ is accurate. By standard generalization arguments, we have with probability at least $1 - \beta/3$,

$$\big|\mathrm{Freq}_H(h) - \mathbb{P}_{S \sim \mathcal{D}^m}\big(G(S) = h\big)\big| \leq \frac{\eta}{8}$$

for every $h \in [K]^{\mathcal{X}}$ as long as $k \geq O(\log(1/\beta)/\eta)$. Conditioned on this event, by accuracy of $\mathrm{HIST}$, with probability $1 - \beta/2$, it produces a list $L$ containing $h^{\star}$ together with a sequence of estimates that are accurate to within an additive error $\eta/8$. Then, $h^{\star}$ appears in $L$ with an estimate $a_{h^{\star}} \geq \eta - \eta/8 - \eta/8 = 3\eta/4$.

Now remove from $L$ every item $h$ with $a_h \leq \frac{3\eta}{4}$. Since every estimate is accurate within $\eta/8$, $h$ appears in $L$ such that $\text{Freq}_H(h) \geq \frac{3\eta}{4} - \frac{\eta}{8} = \frac{5\eta}{8}$. Since sum of frequencies is less than 1, the number of list $L$ should be less than $2/\eta$ (i.e. $|L| \leq 2/\eta$). This list contains $h^\star$ such that $\text{loss}_\mathcal{D}(h^\star) \leq \alpha$. Hence the GENERICLEARNER identifies $h^\star$ with $\text{loss}_\mathcal{D}(h^\star) \leq \alpha/2$ with probability at least $1 - \beta/3$. $\qquad\square$

## C.3 Extension to the Agnostic setting

Theorem 11 showed that online MC learnability continues to imply private MC learnability in the realizable setting. A similar result also holds even when the realizability assumption is violated, which is called *agnostic setting*.

**Corollary 30** (Agnostic setting : Online MC learning implies private MC learning). *Let $\mathcal{H} \subset [K]^\mathcal{X}$ be a MC hypothesis class with $\text{Ldim}(\mathcal{H}) = d$. Let $\epsilon, \delta \in (0,1)$ be privacy parameters and let $\alpha, \beta \in (0, 1/2)$ be accuracy parameters. For $n = O_d\big(\frac{\log(1/\beta\delta)}{\alpha^2\epsilon}\big)$, there exists $(\epsilon, \delta)$-DP learning algorithm such that for every distribution $\mathcal{D}$, given an input sample $S \sim \mathcal{D}^n$, the output hypothesis $f = \mathcal{A}(S)$ satisfies*

$$loss_\mathcal{D}(f) \leq \min_{h \in \mathcal{H}} loss_\mathcal{D}(h) + \alpha$$

*with probability at least $1 - \beta$.*

*Proof.* Alon et al. [5, Theorem 6] propose an algorithm, $\mathcal{A}_{PrivateAgnostic}$, which transforms a private learner in the realizable setting to a private learner that can operate in the agnostic setting. The main idea is based on the standard sub-sampling method, and as a result, the transformed agnostic learner has a larger sample complexity by a factor of $1/\epsilon$. Then Corollary 30 is shown by applying $\mathcal{A}_{PrivateAgnostic}$ to the realizable learner used in Theorem 11. $\qquad\square$

## C.4 Proof of Theorem 15

We complete the proof of Theorem 15. The proof for Condition 4 is given in the main body.

**Theorem 15** (restated). *Let $\mathcal{F} \subset \mathcal{Y}^\mathcal{X}$ be a real-valued function class such that $\text{fat}_\gamma(\mathcal{F}) < \infty$ for every $\gamma > 0$. If one of the following conditions holds, then $\mathcal{F}$ is privately learnable.*

1. *Either $\mathcal{F}$ or $\mathcal{X}$ is finite.*

2. *The range of $\mathcal{F}$ over $\mathcal{X}$ is finite (i.e., $\big|\{f(x) \mid f \in \mathcal{F}, x \in \mathcal{X}\}\big| < \infty$).*

3. *$\mathcal{F}$ has a finite cover with respect to the sup-norm at every scale.*

4. *$\mathcal{F}$ has a finite sequential Pollard Pseudo-dimension.*

*Proof.* 1. If $|\mathcal{F}| < \infty$, then for sample complexity $n = \mathcal{O}(\frac{\log|\mathcal{F}| + \log(1/\beta)}{\alpha\epsilon})$ we directly run the $\epsilon$-DP Generic Private Learner to output with probability at least $1 - \beta$, a hypothesis $\hat{f} \in \mathcal{F}$ such that $\text{loss}_S(\hat{f}) \leq \alpha$. Next, assume that $\mathcal{X}$ is finite. The finiteness of $\mathcal{X}$ does not imply finite $|\mathcal{F}|$ because $\mathcal{Y}$ is continuous, but we can discretize $\mathcal{F}$ at some scale $\gamma$, which gives us a finite MC hypothesis class $[\mathcal{F}]_\gamma$. It is private-learnable by $\epsilon$-DP Generic Private Learner, and then the original class $\mathcal{F}$ is also privately-learnable within accuracy $\gamma$.

2. Observe that this regression problem is essentially a MC problem. Furthermore, $\text{Ldim}(\mathcal{F})$ by considering it as a MC problem is bounded above by $\text{fat}_\gamma(\mathcal{F})$, where $\gamma$ is the minimal gap between consecutive values in the range of $\mathcal{F}$ over $\mathcal{X}$. This means that $\text{Ldim}(\mathcal{F})$ is finite, and hence by the argument of Section 5.1, $\mathcal{F}$ is privately learnable.

3. Given an accuracy $\alpha$, $\mathcal{F}$ has $n$ finite covers with a radius $r < \alpha$. We construct a set of representative function as $\mathcal{F}' = \{f_1, \cdots, f_n\} \subset \mathcal{F}$ by arbitrarily choosing a representative $f_i$ from the $i$-th cover, and then run $\epsilon$-DP Generic Private Learner on $\mathcal{F}'$ to output a hypothesis $\hat{f} \in \mathcal{F}$ with a small population loss. $\qquad\square$

## Footnotes

[2] A subset of the universe is homogeneous if all of its $t$-subsets have the same color.