[Reviews · NeurIPS 2020]

Review 1

Summary and Contributions: Recent paper established equivalence between private learning and online learning in the setting of binary classification. The current paper investigate this connection in the context of multi-class classification and regression. For multi-class classification, the authors show that the equivalence remains true. For regression, they show that one direction remains true, namely, that private learning implies online learning. The other direction remains open, but the authors do show certain hurdles in current proof techniques. To establish their result, the authors define and investigate a new form of Littlestone dimension, the accommodate tolerance.

Strengths: This is a solid extension of recent work

Weaknesses: I don't see a particular weakness

Correctness: yes

Clarity: yes

Relation to Prior Work: yes

Reproducibility: Yes

Additional Feedback:


Review 2

Summary and Contributions: This submission considers the connection between private learning and online learning. Recently, differential privacy has emerged as a widely-used standard for releasing aggregate statistics about a dataset while protecting the privacy of individuals in the dataset. Therefore, a fundamental question has been to explore PAC learnability in this private setting. Meanwhile, online learning has been an important model in statistical learning theory--in this setting, a learner and an adversary interact and, in each time step, the learner receives an example from the adversary and has to predict a label, after which he subsequently observes a loss. The goal is to minimize regret or cumulative loss among the sequence of observed examples. Private learnability and online learnability were recently shown to be equivalent for binary classification tasks (and are characterized by concept classes with finite Littlestone dimension). The authors' submission seeks to investigate whether the equivalence applies in the case of multi-class classification and regression. To this extent, the authors introduce a modified version of Littlestone dimension that extends beyond binary classification and makes use of a tolerance parameter. The authors are able to show that, as in the binary case, online learnability and DP PAC learnability are equivalent for both multi-class classification and regression. Moreover, for the case of multi-class classification, they show that the converse is also true, i.e., DP PAC learnability implies online learnability.

Strengths: The problem is an interesting one and is a well-motivated extension of recent work of Bun et al. (FOCS '20). The authors manage to extend the results to multi-class classification, which is important class of learning tasks. As both differential privacy (in the context of learning tasks) and statistical learning are of great interest to the NeurIPS community, this work falls within scope. As this work is solely theoretical (no experiments), the theoretical grounding is quite strong.

Weaknesses: The author leave open the question of equivalence for regression and don't seem to offer much intuition on whether we expect a similar result to hold (as in the case of classification). It is unclear whether the limitation is mainly in the proof techniques of this work or in correctness of the statement. The variant of Littlestone dimension that the authors propose isn't conceptually too difficult (it is a natural extension of the binary case, with some thresholding magic).

Correctness: As far as I can tell, yes. But I have not verified proofs in the appendix.

Clarity: I found the paper well-written and easy to follow.

Relation to Prior Work: Yes, the work talks about Alon et al. and Bun et al., which are the main works in this space.

Reproducibility: Yes

Additional Feedback: N/A


Review 3

Summary and Contributions: Update following the the author response: I thank the authors for the clarifications. This paper explores the connection between learning with approximate privacy and online learning for multi-class and regression problems. It follows prior work that showed an equivalence between approximate privacy and online learning for binary classification. This paper has two main results: (1) Multi-class: Consider a hypothesis class H of functions from X to Y where Y is finite. Then, H is PAC learnable with respect to the 0-1 loss in the online setting with if and only if it is learnable with approximate privacy in the standard stochastic batch setting. (2) Regression: Consider a class of functions from X to Y, where Y is a bounded subset of the reals. If H is learnable with respect to the absolute value difference (L1) loss with approximate privacy, then it is online learnable. The second direction (online to private) requires some additional assumptions. * Notice that the reductions can result in an super-exponential blow-up of the sample complexity, as is the case for binary classification. The paper answers a fundamental question, as it strengthens the connection between two well studied learning models (privacy and online learning). Since multi-class and regression are fundamental learning settings, this will be of interest to the community. In terms of the theoretical novelty, it does not seem that the proof requires significant new ideas compared to prior work: first the proof for the multi-class setting seems to follow through, with small adjustments. And secondly, the regression theorems follow from the multi-class theorem by discretization in a simple manner. However, the fact that the authors managed to avoid complications and present simple proofs can be taken as an advantage. For conclusion: the paper answers a fundamental question that is relevant to the NeurIPS community. The proof did not seem to overcome any significant barrier. However, due to the importance of the result and the clarity of the presentation, I vote for acceptance.

Strengths: - Fundamental question - Simple and clear solution

Weaknesses: No significant theoretical barrier was overcome

Correctness: The claims seem to be theoretically correct

Clarity: The paper is mostly well written

Relation to Prior Work: The relation to prior work is clearly discussed

Reproducibility: Yes

Additional Feedback: - Algorithm 2 is not sufficiently formal, hence it is hard to follow the proof sketch. - Was a reduction between multiclass and regression used in prior work generally in ML? It would be worth mentioning this.


Review 4

Summary and Contributions: The paper looks at the connections between online and private learnability in multi-class classification and regression with absolute loss. The authors show that - Private MC and regression learnability implies online learnability - Online MC learnability implies private MC learnability - They explain some problems that occur when trying to prove that online regression learnability implies private regression learnability, and give some sufficient conditions for private regression learnability.

Strengths: This paper was very pleasant to read and nicely explained. The authors look at some very recent results from DP theory and how they may be generalized to settings besides binary classification. To the best of my knowledge, these are the first results of this type.

Weaknesses: I don’t see any major weaknesses. Of course it would be more groundbreaking to fully resolve the regression setting with absolute loss, instead of just part of it along with MC classification. But I think given the conference page limit, there is enough novelty here to warrant publication, and hopefully these results fuel more thought in this direction.

Correctness: Update after author response: Thanks to the authors for the clear and helpful response! My concerns are fully addressed. ================================== As far as I can tell, claims and method are correct, save for two unclear parts that need to be resolved: - In Sec 2.3 the authors state that after each round, the learner observes the loss of the given prediction. In the binary setting, this is equivalent to observing the correct label, but not in the MC and regression settings. In the MC setting, this would be online learning with bandit information, which is less straightforward to characterize and afaik only fully analyzed for the realizable case in [10]. Did the authors mean to let the learner observe the correct label after each round? If so, Sec 2.3 should be updated. If not, the authors should explain how this choice affects subsequent results. - In Sec 5.1 the main Theorem is Theorem 11, which proves that for hypothesis classes with finite Littlestone dimension, there exists a PAC DP learner for every realizable distribution. I am missing a clear discussion of how to extend this result to the non-realizable case. There is some discussion on extending SOA to non-realizable sequences in the appendix, but it’s fairly hidden and not discussed anywhere in the main text. At the same time, Definition 1 allows a general distribution over X \times Y, so I would assume the agnostic setting, but bounds the absolute loss of the output hypothesis instead of the relative loss, which implies realizability. I would ask the authors to make this clearer in the text, and to explain in the rebuttal/discussion which setting they are considering and what the implications are in the different parts of the paper.

Clarity: The paper is very clear and pleasant to read. Sec 5 is very dense, as evidenced by the amount of supplementary material on this Section.

Relation to Prior Work: Yes

Reproducibility: Yes

Additional Feedback: I suggest that when proving a Theorem in the appendix/supplementary material, repeat the Theorem so that the reader does not have to go back and forth. On my first read, I felt that the definition of Pollard pseudo-dimension was a bit shoehorned into Sec 2.3. Perhaps it would be more clear to only define Littlestone and fat-shattering dimensions in Sec 2.3 and introduce the Pollard pseudo-dimension in Sec 5.2. (This is just a suggestion.) Also, I didn’t see an explicit statement in Sec 2.3 that H is MC-online-learnable iff Ldim is finite. This should be added for clarity. Typo: Line 33 “it was unknown that what”

[Author Response · NeurIPS 2020]

We really appreciate the time and expertise you have invested in these reviews. We wish to express our appreciation for your in-depth comments, suggestions, and corrections, which will greatly improve the manuscript. Please see below for our responses to individual questions and comments.

**Reviewer #1**

We thank for your positive feedback.

**Reviewer #4**

*The limitation of proving the equivalence in regression:* The major hurdle that we explained in the paper is specific to existing proof techniques. That is to say, there is no obvious way to extend the current proof in the classification setting to the regression setting. It is still possible that there is another method to prove the result for regression. To the best of our knowledge, there is neither positive nor negative evidence whether online regression learnability implies private regression learnability. We will make it clearer in the final version that the limitation stems from currently known proof techniques.

**Reviewer #5**

*Presentation of Algorithm 2:* We will make Algorithm 2 more formal and make the proof of Theorem 8 more readable.

*Reduction from regression to classification:* There are papers that use regression models in multi-class classification (e.g., see Rakesh, K., & Suganthan, P. N. (2017). An ensemble of kernel ridge regression for multi-class classification. or Yang, Z., Deng, N., & Tian, Y. (2005). A multi-class classification algorithm based on ordinal regression machine.) However, we are not aware of any previous work that studies regression learnability by transforming the problem into a discretized classification problem. Furthermore, our work is the first one that proposes the Littlestone dimension with tolerance, which is the main key to bridge regression and classification learnability. We will clarify these points in the final version.

**Reviewer #6**

*Bandit or full information feedback:* We considered the full information setting in that the learner receives the true label information after making a prediction. Thanks for raising this issue, and we will update Section 2.3 to clarify that the learner gets full information feedback.

*Realizable or agnostic settinig:* For the sake of clear presentation, we only discussed the realizable setting in the paper. Alon et al. (2019) only consider the realizable setting while Bun et al. (2020) discuss the extension to the agnostic setting. In fact, at least for the direction that online learnability implies private learnability, it is not hard to extend the argument to the agnostic setting. For example, in the manuscript, "Closure Properties for Private Classification and Online Prediction," Alon et al. (2020) show that private learning implies private agnostic learning (see their Theorem 2.4). We will make this clearer in the main text, and introduce the aforementioned theorem in the appendix to make the story complete.

*Additonal comments:* Thanks for your suggestions for improving presentation. We will address them in the final version.

[Meta-Review · NeurIPS 2020]

This is overall a good work that technically relies on previous work. It contributes to our understanding of the relation between online learnability and privacy.